

# Off-line and on-line optical monitoring of microalgal growth

Hugo-Enrique Lazcano-Hernández[1], Gabriela Aguilar[2], Gabriela Antonia Dzul-Cetz[2], Rodrigo Patiño[2] and Javier Arellano-Verdejo[3]

[1] Cátedras CONACYT-El Colegio de la Frontera Sur, Chetumal, Quintana Roo, México
[2] Departamento de Física Aplicada, Cinvestav Unidad Mérida, Mérida, Yucatán, México
[3] Estación para la Recepción de Información Satelital ERIS-Chetumal, El Colegio de la Frontera Sur, Chetumal, Quintana Roo, México

## ABSTRACT

The growth of *Chlamydomonas reinhardtii* microalgae cultures was successfully monitored, using classic off-line optical techniques (optical density and fluorescence) and on-line analysis of digital images. In this study, we found that the chlorophyll fluorescence ratio $F_{685}/F_{740}$ has a linear correlation with the logarithmic concentration of microalgae. By using digital images, the biomass concentration correlated with the luminosity of the images through an exponential equation and the length of penetration of a super luminescent blue beam ($\lambda$ = 440 nm) through an inversely proportional function. The outcomes of this study are useful to monitor both research and industrial microalgae cultures.

Corresponding authors
Rodrigo Patiño,
rodrigo.patino@cinvestav.mx
Javier Arellano-Verdejo,
javier.arellano@mail.ecosur.mx

## INTRODUCTION

Photosynthesis is a biophotonic mechanism by which green plants, cyanobacteria, and algae produce their own food by transforming a fraction of solar energy to biochemical energy. This is the foundation of life on Earth. Photosynthesis occurs in the chloroplasts, which are cell organelles that contain photosynthetic pigments (chlorophyll (Chl) a, Chl b, carotenoids, etc.). They absorb light and use it to drive photosynthetic light reactions and associated electron transport reactions to reduce $CO_2$ and oxidize $H_2O$ in the Calvin cycle (*Allen, 1992*). The net result of photosynthesis is the production of carbohydrates and the release of molecular oxygen to the atmosphere. Environmental factors such as temperature, irradiance, humidity, and salinity are known to affect photosynthesis (*Rym, 2012*).

Microalgae cultivation has been widely studied due to its potential as a source of food, biofuel, and various bioactive compounds (*Pulz & Gross, 2004*; *Spolaore et al., 2006*; *Mata, Martins & Caetano, 2010*; *Niccolai et al., 2019*). These are useful for important processes such as residual water cleaning, $CO_2$ capture, and $H_2$ synthesis (*Wang et al., 2008*; *Abdel-Raouf, Al-Homaidan & Ibraheem, 2012*; *Show et al., 2019*). All these are valuable contributors to the balance and growth of human activity on a global scale

(*Gupta, Lee & Choi, 2015*). *Havlik, Scheper & Reardon (2016)* compiled a wide review of on-line and off-line technologies to monitor physicochemical and biological parameters of microalgae. There are also several photobioreactor (PBR) models to predict growth (*Pulz, 2001*; *Carvalho, Meireles & Malcata, 2006*; *Xu et al., 2009*). However, actual measurements are required to monitor and optimize the algae growth process. Disadvantages of sampling include the potential for contaminating the culture, disturbing the algae's physiological state or modifying the volume of the medium. Another challenge for real-time measurements is the wide range of concentrations encountered in microalgal cultures. The concentration routinely increases by up to three orders of magnitude, preventing direct measurement by most analytical methods (*Antal et al., 2019*). Therefore, it is currently a challenge to implement non-invasive real-time methodologies for monitoring microalgal cultivation conditions and photosynthetic parameters (*Antal et al., 2019*).

Chlorophyll molecules are organized into two different light systems called Photosystem I (PSI) and Photosystem II (PSII). Both are spatially separated in the chloroplasts' thykaloid membranes (*Breijo, Caselles & Siurana, 2006*). Every photosystem contains an antenna light-harvesting complex (LHC) and central Chl molecules. The photosystems differ from each other in their proportions of Chl a, Chl b, the characteristics of their reaction centers, and the electron carriers involved in their processes. In PSI, the reactive center is called P700 and is formed by two Chl a molecules that are attached to each other. PSII also contains a reactive center called P680 which is formed by two attached Chl a molecules. The nomenclature is associated with the maximum wavelength ($\lambda$) absorption of both PSI and PSII: $\lambda = 700$ nm and $\lambda = 680$ nm, respectively (*Gouveia-Neto et al., 2011*). The maximum fluorescence wavelength may vary according to the origin and kind of Chl, the culture medium, the environmental conditions, and the measurement equipment. For example, at room temperature, Chl a fluorescence around $\lambda = 685$ nm is largely emitted by PSII antenna, and fluorescence around $\lambda = 740$ nm is emitted by PSI antenna (*Krause & Weis, 1984*; *Roháček, Soukupová & Barták, 2008*; *Gouveia-Neto et al., 2011*). In the fluorescence emission spectra of healthy, suspension-diluted thylakoid membranes or isolated chloroplasts, a sharp peak around $\lambda = 685$ nm with a broad shoulder at about $\lambda = 740$ nm has been observed (*Krause & Weis, 1991*). Although isolated Chl b dissolved in an organic solvent exhibits fluorescence, this does not happen with in vivo cultures because the excitation energy is transferred completely to Chl a (*Gouveia-Neto et al., 2011*).

The main function of the LHC is to transfer excitation energy to the photosynthetic reaction centers, where photochemical reactions take place. However, a part of the absorbed light energy is dissipated as heat or emitted as fluorescence (*Misra, Misra & Singh, 2012*). In other words, to return to the ground state, the excited Chl molecule undergoes one of three processes including: (i) driving photochemical reactions (photosynthesis), (ii) dissipating as heat (thermal de-excitation), or (iii) being re-emitted as light (fluorescence). These processes occur in competition so that any increase in the efficiency of one will result in a decrease in the yield of the other two. Chl fluorescence is an intrinsic signal emitted by plants, algae, and cyanobacteria that can be employed to

monitor changes in their physiological state, such as in the photosynthetic apparatus, developmental processes, state of health, stress events and stress tolerance. It can also be used to detect diseases or nutrient deficiency (*Gouveia-Neto et al., 2011*; *Hák, Lichtenthaler & Rinderle, 1990*). Hence, by measuring the yield of Chl fluorescence, information about changes in the efficiency of photosynthesis and heat dissipation can be obtained (*Maxwell & Johnson, 2000*; *Krause & Weis, 1984*). Therefore, the simultaneous measuring of Chl fluorescence at $\lambda = 685$ nm ($F_{685}$) and $\lambda = 740$ nm ($F_{740}$) allows for an approximate non-destructive determination of Chl content using the Chl ratio ($F_{685}/F_{740}$) (*Hák, Lichtenthaler & Rinderle, 1990*).

Depending on the type of study and the suitability of the photosynthetic system, different fluorescence techniques have been used (*Mauzerall, 1972*; *Olson, Chekalyuk & Sosik, 1996*; *Kolber, Prášil & Falkowski, 1998*; *Gorbunov & Falkowski, 2004*; *Johnson, 2004*; *Chekalyuk & Hafez, 2008*). Presently, two Chl fluorescence approaches are successfully used to monitor photosynthetic efficiency in microalgae mass cultures: rapid fluorescence induction and the saturation-pulse method (*Masojídek et al., 2011*). Regarding outdoor algae cultures, specific fluorimeters have been used. The pulse amplitude modulation (PAM) fluorimeter provides rapid light responses curves of PSII. The dual PAM fluorimeter estimates PSI and PSII yields, and the Induction Kinetics fluorimeter measures fluorescence induction curves (*Sukenik et al., 2009*; *Kromkamp et al., 2009*; *Masojídek, Vonshak & Torzillo, 2010*). On the other hand, the dry weight of microalgae correlated with the optical density (OD) of the algae cultures (*Del Campo et al., 2014*). Limitations of the OD technique, as well as its applications to other species and another culture medium, were discussed by *Griffiths et al. (2011)*. However, real-time non-invasive methodologies are still needed to monitor microalgae culture growth conditions. Therefore, two room temperature fluorescence methods to measure *Chlamydomonas reinhardtii* culture growth are proposed in this study. The first is an analytic off-line optical technique and the second involves on-line digital image analysis. The study of *C. reinhardtii* here is relevant since it is considered one of the most promising eukaryotic $H_2$ producers (*Torzillo et al., 2015*). It is possible to apply the proposed methods to other species but results will vary according to specific strain features, the culture medium, PBR geometry, light sources, and temperature. However, the methods are simple and easy to implement and assess.

This project aimed at identifying alternative optical techniques for monitoring microalgal growth. Our hypotheses were that *C. reinhardtii* cell culture concentration correlates with (1) the off-line Chl fluorescence ratio $F_{685}/F_{740}$ and (2) the on-line image culture color, and (3) the on-line image culture fluorescence.

## MATERIALS AND METHODS

### Microalgae cultures

*Chlamydomonas reinhardtii* (CC-124) microalgae were purchased from the Chlamydomonas Resource Center (USA) and were grown photoautotrophically in a Sueoka medium (*Sueoka, 1960*). Growth conditions included continuous air bubbling (1 VVM = 1 L − air/min/L − medium) under controlled room temperature conditions

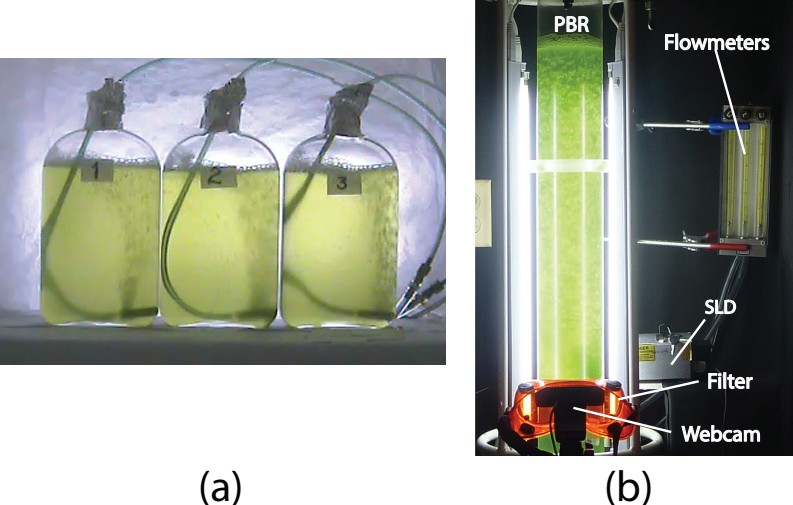

**Figure 1** (A) The three Roux bottles for the off-line experiments (A and B), and (B) the column photobioreactor (PBR) for the on-line experiments (1–5), Super Luminescent Diode (SLD); see "Materials and Methods."

(298 ± 2) K. Experiments were performed in two series. The first series was conducted for the off-line OD and fluorimetry measurements (Experiments A and B). The second series was conducted for the on-line techniques that used digital images (Experiments 1–5). A portable spectrometer (EPP2000; StellarNet, Tampa, FL, USA) was used to measure the OD, fluorescence and color of the microalgae cultures, with a detection range between 200 and 850 nm. Spectrawiz software (OS V5.0©2011; StellarNet) was used for spectrometer measurements. In order to monitor cell growth, OD measurements at $\lambda =$ 640 nm were taken. Each measurement was repeated three times for every sample.

For the off-line experiments, three $1 - L$ Roux culture bottles were used, each with 0.9 L of algae culture and continuous air bubbling (one L/min), under controlled room temperature conditions (Fig. 1A). The cultures were illuminated continuously with two fluorescent lamps (Philips, F20T12/D 20W). To assess the microalgae growth kinetics, three samples of three mL each were taken from every bottle every 12 h for 5 days. Additionally, a final control measurement was taken on the 7th day. Triplicate OD and fluorescence measurements were taken for every sample. Two experiments were performed with the only difference between them being the initial microalgae concentration ($x_0$). These were labeled Experiment A ($x_0 = 34 \pm 2$ mg/L), and Experiment B ($x_0 = 42 \pm 2$ mg/L). In summary, a total of 270 measurements were made for every off-line experiment and the number of measurements provided statistical significance to this study.

For the on-line experiments, a $3 - L$ column PBR was used (Fig. 1B). A transparent acrylic tube with a thickness of 25 mm and an inner diameter of 95 mm was used to build the $95 - cm$ length PBR. Four fluorescent lamps (T4 6500K20W; Tecno Lite, China) were used to illuminate the center of the column with around 100 μmol photons m$^{-2}$ s$^{-1}$. To avoid external light, the PBR was placed inside a dark cabin. Five experiments were performed, each with 5 days of sampling. These included three experiments with
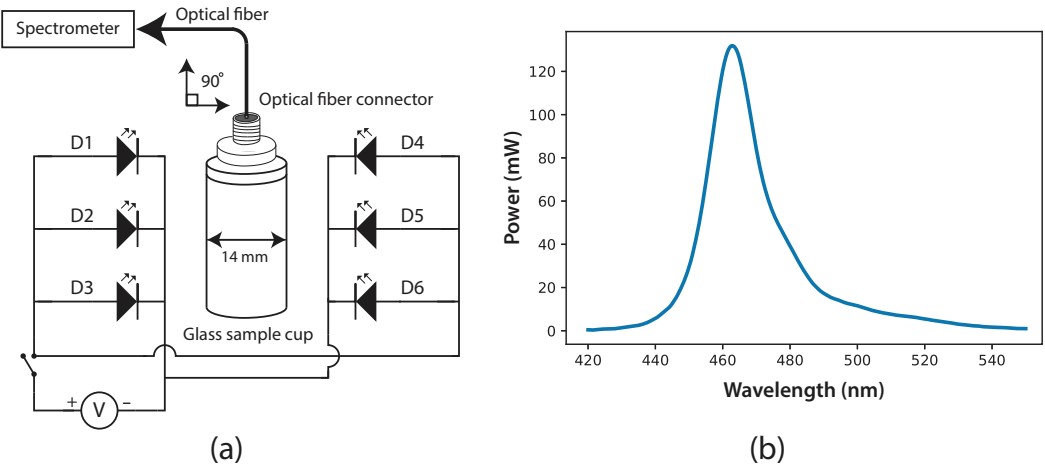

**Figure 2 (A) Fluorescence cabin diagram, with optical fiber connector and glass sample cuvette; six Ultra Blue LEDs were installed as exciting source.** (B) Spectrum of the exciting radiation source (Ultra Blue LEDs, $\lambda_{max}$ = 464 nm).

continuous illumination (Experiments 1, 2, and 3), and two experiments with 12-h light/12-h dark cycles (Experiments 4 and 5). Both off-line samples and digital images were taken every 6 or 12 h. A webcam (Carl Zeiss Tessar HD 1080 p; Logitech, China) connected to a PC was used to capture the PBR digital images. For the color analysis, the images were captured with two fluorescent lamps located behind the PBR. These lamps were turned on while the other two lamps remained turned off. The fluorescent images, were captured in darkness and only the culture was illuminated with a blue beam from a super luminescent diode (maximal wavelength of 440 nm) and a light filter (LSR-GARD ARGON, model 2204) with a protection range within 190–520 nm. In order to measure fluorescence after a dark-adapted period, at least 15 min of darkness was provided before fluorescent stimulation. The images were analyzed in a LCD screen with an integrating Cube (IC2; StellarNet) and portable spectrometer. The International Commission on Illumination (CIELAB) scale was used for color measurements. To achieve a homogeneous representation of the color of every image, five measurements were taken from different regions of the PBR.

## Fluorescence cabin

The experimental system to measure fluorescence was composed of two main parts: a fluorescence cabin and the portable spectrometer. Based on the spectrometer, the fluorescence cabin was designed, manufactured, and coupled to the system using an optical fiber (F400; StellarNet). Through this fiber, light was guided from the sample cuvette to the spectrometer detector. The fluorescence cabin configuration is shown in Fig. 2A. The main components are a dark cabin, a cylindrical (14 mm i.d.) glass sample cuvette (four mL), six light emission diodes (LED), feed and switching electronic circuits, an AC/DC electric current converter (output 5.4 V), and a multi-modal optical fiber connector. The dark cabin is a space where light does not come in from external sources. Inside the dark cabin, a base was used to fix the cuvette in a normal position (at 90°)

relative to the floor. As exciting radiation, six LEDs (Ultra Blue; Steren, Shangai, China) were placed parallel to the floor, three to the right side of the sample cuvette and three to the left. This configuration ensured homogeneous illumination conditions. In Fig. 2B, the spectrum of the six LEDs is shown, with a maximum wavelength emission at around $\lambda = 464$ nm, luminosity of 7 cd and 400 mW as maximum power.

To improve the quality of fluorescence measurement, the optical fiber was positioned on top of the glass sample cuvette, at 90° relative to the LEDs' radiation. Fluorescence traditionally is measured through the sample cuvette wall, but our fluorescence measurements were taken at the uncovered top of the cuvette, thus reducing losses from reflection and refraction in the interface of the cuvette. The cuvette used for the samples was a four-mL glass cylinder, but with this fluorescence system, the cuvette's geometry is not important. Light information was processed and digitized using the spectrometer SpectraWiz software. Before measurements, the reference blank was defined by setting the spectrometer with the Sueoka medium, then the fluorescence measurement was performed on the microalgae sample. The direct fluorescence of *C. reinhardtii* culture samples at room temperature was measured successfully for 7 days with this experimental setup. The ratio of fluorescence intensity between $\lambda = 685$ and $\lambda = 740$ nm was calculated.

## The $F_{685}/F_{740}$ fluorescence ratio

At low Chl concentrations, fluorescence emissions increase with increasing amounts of Chl. At higher concentrations, the increase of fluorescence with the increment of Chl is mainly detected around 740 nm. For in vivo cultures, fluorescence emission at 740 nm is favored and fluorescence emission at 685 nm is not favored. This is due to the following factors: (i) the re-absorption of photons from the fluorescence emitted by neighboring molecules, (ii) light interference between the short (685 nm) and long wavelengths (740 nm), and (iii) the increment of Chl (the new Chl molecules preferentially absorb energy at 685 nm) (*Gouveia-Neto et al., 2011*).

There is a good inverse correlation between photochemistry and Chl fluorescence. The ratio of fluorescence intensity between maximal wavelengths ($F_{685}/F_{740}$) is influenced by photosynthetic activity. In mature microalgae cultures, the chloroplast structure, $CO_2$ uptake rate, carbon metabolism, etc., are better than in younger cells. Higher $F_{685}/F_{740}$ values signal young cultures or cultures with a photosynthetic apparatus that isn't developing. Low values of this rate indicate mature cultures with a fully developed photosynthetic apparatus. In other words, a decrement in $F_{685}/F_{740}$ values is indicative of increased photosynthetic activity. Measured by induction fluorescence, $F_{685}/F_{740}$ exhibits a curvilinear relationship with cell concentration ($x$). This relationship is successfully expressed in Eq. (1), where $c$ and $d$ are constants (*Hák, Lichtenthaler & Rinderle, 1990*):

$$\frac{F_{685}}{F_{740}} = cx^{-d} \tag{1}$$

This technique has been applied to all kinds of leaves, chloroplast suspensions, and acetone extracts of photosynthetic pigments. Our study demonstrated that this technique is also applicable to microalgae cultures.

**Table 1 The Gompertz specific growth rate of microalgae in seven experiments, with the corresponding correlation values.** The mean value and the standard deviation are reported for the cultures with continuous illumination (Experiments A and B in the Roux bottles, and 1, 2, and 3 in the PBR) and with light/dark cycles (Experiments 4 and 5 in the PBR).

| Reactor | Experiments | $\mu$ (day$^{-1}$) | $R^2$ |
|---|---|---|---|
| Roux bottles | AI | 0.5472 | 0.9981 |
| | AII | 0.5856 | 0.9989 |
| | AIII | 0.5424 | 0.9889 |
| | | 0.56 ± 0.02 | |
| | BI | 0.3360 | 0.9843 |
| | BII | 0.4032 | 0.9965 |
| | BIII | 0.3696 | 0.9944 |
| | | 0.37 ± 0.03 | |
| PBR | 1 | 0.7824 | 0.9954 |
| | 2 | 0.6720 | 0.9771 |
| | 3 | 0.6192 | 0.9656 |
| | | 0.69 ± 0.08 | |
| | 4 | 0.3384 | 0.9180 |
| | 5 | 0.4776 | 0.9312 |
| | | 0.41 ± 0.10 | |

# RESULTS AND DISCUSSION

## Microbial growth

It has been established that the Gompertz model represents *C. reinhardtii* growth better than the classical Monod model (*Del Campo et al., 2014*). Actually, the Monod and Gompertz models can be seen as particular cases of a more universal growth model (*Castorina, Delsanto & Guiot, 2006*). For the experiments in the Roux bottles (A and B), as well as in the PBR (1–5), the Gompertz specific growth rate is reported in Table 1. For the PBR experiments, it was possible to observe that the Gompertz model is best fitted when the light regime is continuous (see $R^2$ for Experiments 1–3) than when light/dark cycles are performed (Experiments 4–5). Moreover, the specific growth rate is favored when the PBR is used. This stands in contrast to the cultures in the Roux bottles.

## Fluorescence measurements

Variations in fluorescence intensity were successfully measured according to the increment in *C. reinhardtii* concentration. Some selected spectra are shown in Fig. 3 for experiment A. The fluorescence dataset for experiments A and B are shown in File S1. In all cases, Chl fluorescence exhibits a peak around $\lambda = 685$ nm and a broad shoulder around $\lambda = 740$ nm. This is a general observation at room temperature for Chl a (*Gouveia-Neto et al., 2011*; *Krause & Weis, 1984*). The fluorescence around $\lambda = 685$ nm is attributed to the PSII antenna, and the fluorescence around $\lambda = 740$ nm is due to the PSI antenna (*Gouveia-Neto et al., 2011*). The fluorescence signal/noise ratio in the measurements was around 7 at $\lambda = 740$ nm and 14 at $\lambda = 685$ nm.

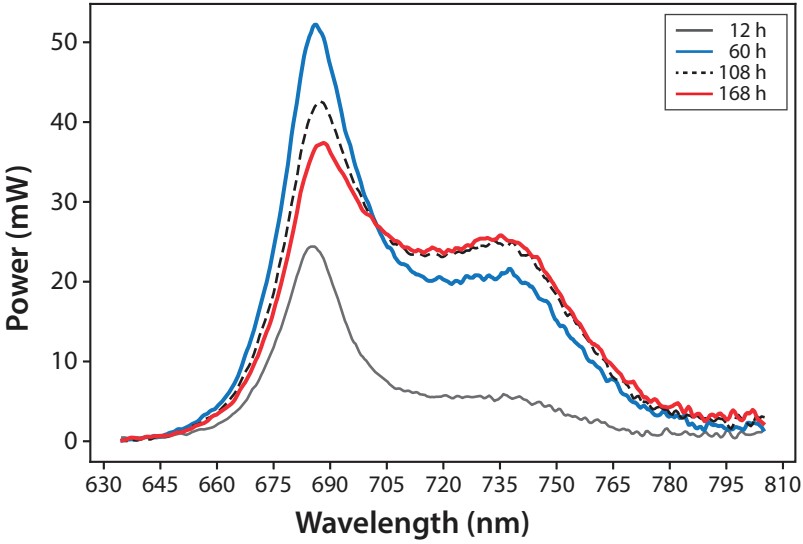

**Figure 3 Microalgae fluorescence emission spectra (298 ± 2 K): evolution over 7 days (168 h).** Experiment A, $x_0 = (34 \pm 2)$ mg/L.   

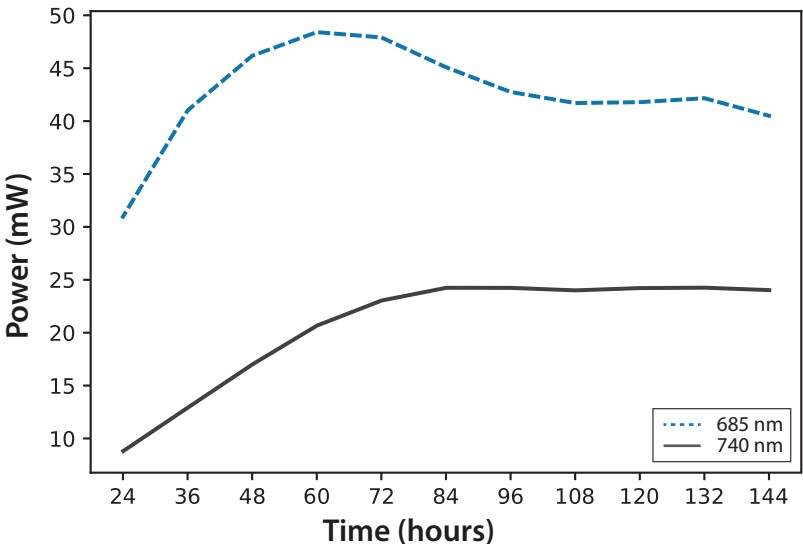

**Figure 4 Experiment A: *C. reinhardtii* fluorescence evolution trend at $\lambda = 685$ nm and $\lambda = 740$ nm.**

  In Fig. 4, fluorescence evolution at $\lambda = 685$ nm and at $\lambda = 740$ nm is shown for all bottles of Experiments A. The plots include trend lines. In both Experiments A and B, the maximum fluorescence intensity at $\lambda = 685$ nm occurred between 48 and 60 h. After that time, fluorescence decreased. Regarding fluorescence at $\lambda = 740$ nm, in Experiment A, the maximum value occurred at 72 h in all cultures. In Experiment B, the maximum value happened between 96 and 108 h. For both experiments, the maximum fluorescence at $\lambda = 685$ nm occurred before that of $\lambda = 740$ nm. Maximum fluorescence at $\lambda = 740$ nm occurred at the highest algae concentrations. In general, the results were consistent with those described in the literature for fluorescence in plants (*Gouveia-Neto et al., 2011*).
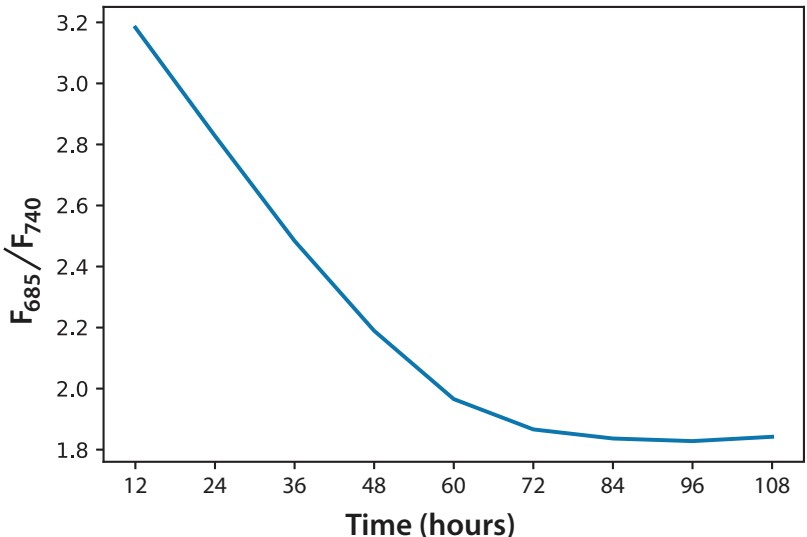

**Figure 5 Fluorescence ratio ($F_{685}/F_{740}$) trend for *C. reinhardtii* for all experiments.**

Namely, at low Chl concentrations, fluorescence emissions increased with increasing Chl concentration. At higher concentrations, the increase of fluorescence with the increment of Chl was detected only around $\lambda$ = 740 nm.

Regarding the $F_{685}/F_{740}$ ratio, we observed a similar behavior in every culture of both experiments, regardless of the initial concentration. For that reason, Fig. 5 shows the average of the three cultures for each of the two experiments over 5 days. Over time, the $F_{685}/F_{740}$ fluorescence ratio decreased, meaning that the photosynthetic processes had improved. The same trend has been reported for green leaves in plants, as stated in Eq. (1) (*Hák, Lichtenthaler & Rinderle, 1990*).

Based on the information in Fig. 5, and considering the data of 168 h as the minimum possible value (aged cultures), we figured that cultures in both experiments reached around 70% maturity at 96 h. Therefore, every culture evolved successfully and the conditions were appropriate to grow the microalgae and keep them in a good state of health. In addition, the cultures that reached the lowest $F_{685}/F_{740}$ values, that is, the highest photosynthetic activity, were those of Experiment A, which started with the lowest initial concentration. Moreover, after 72 h, these values did not change significantly. This is the moment when illumination may not be enough for the culture because cell concentration reduces the passage of light. Equation (2) expresses a very useful linear correlation between the logarithmic concentration of microalgae and the $F_{685}/F_{740}$ ratio through time (Fig. 6):

$$\ln\left(\frac{x}{x_0}\right) = 3.27 - 0.7084\,(F_{685}/F_{740}) \tag{2}$$

## Digital images

When the PBR was used, digital images of the cultures were taken to monitor the change in color and to measure the penetration of a fluorescent beam during microalgae growth.

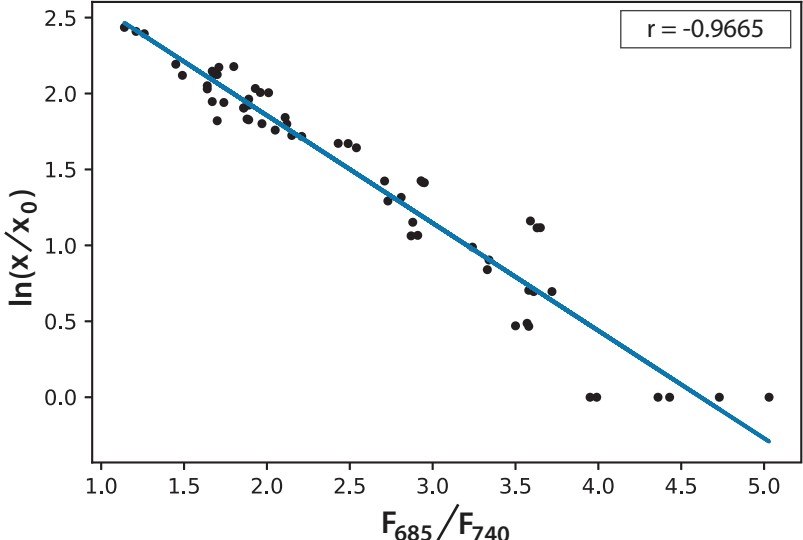

**Figure 6 Linear correlation for the logarithmic microalgae concentration and the fluorescence ratio** $F_{685}/F_{740}$ **(**$r = -0.966578$**).**

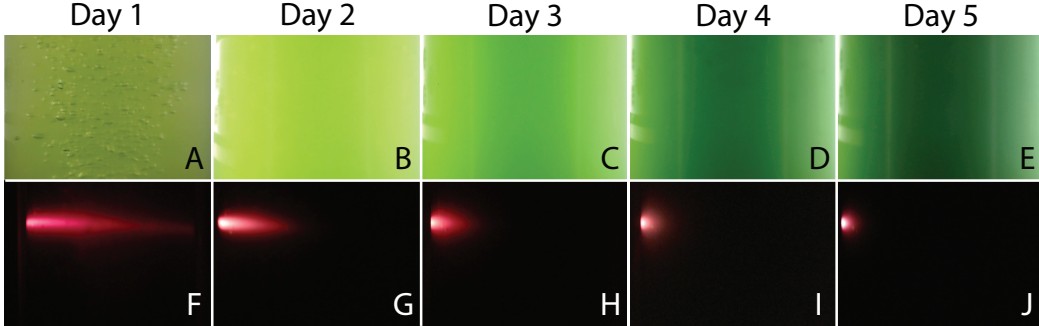

**Figure 7 Representative images of the PBR captured every day during an experiment.** Color assessment (A–E) and fluorescent beam penetration (F–J).

Figure 7 shows a selection of images illustrating a typical experiment. We observed that cultures got darker with time due to the increase of biomass concentration in the PBR preventing the passage of light throughout the reactor. For the fluorescence measurements, the flashes due to the blue super-luminescent diode were filtered in order to measure only fluorescence light contribution. This contribution diminishes with time due to a shadow effect produced by cells as the microalgae concentration gets denser.

International Commission on Illumination measurements include three values to characterize the color of a sample: L is the luminosity, the parameter "a" represents colors from green to red, and the parameter "b" represents colors from blue to yellow. Both parameters "a" and "b" remained almost constant throughout the experiment during microalgae growth. This means that, technically, color does not change. This is expected since the photosynthetic pigments are always the same and luminosity is what more importantly diminishes during cell growth since cells deflect or shadow light sources. Figure 8 shows the logarithmic correlation between microalgae concentration $x$, and

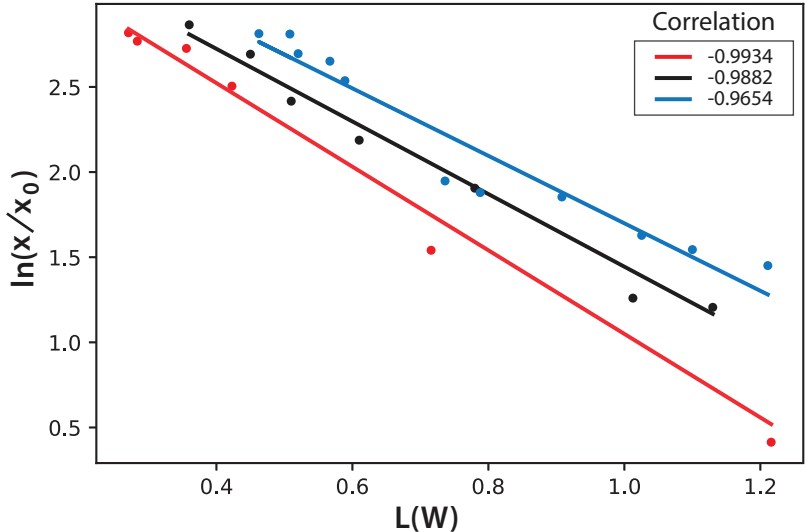

**Figure 8 Logarithmic dependence of cell concentration with the luminosity (L) value in the CIELAB scale of colors for the PBR experiments with continuous illumination.** The colors correspond to experimental values: black (Exp. 1), red (Exp. 2), and blue (Exp. 3); the lines correspond to the mean-squares correlations.                 

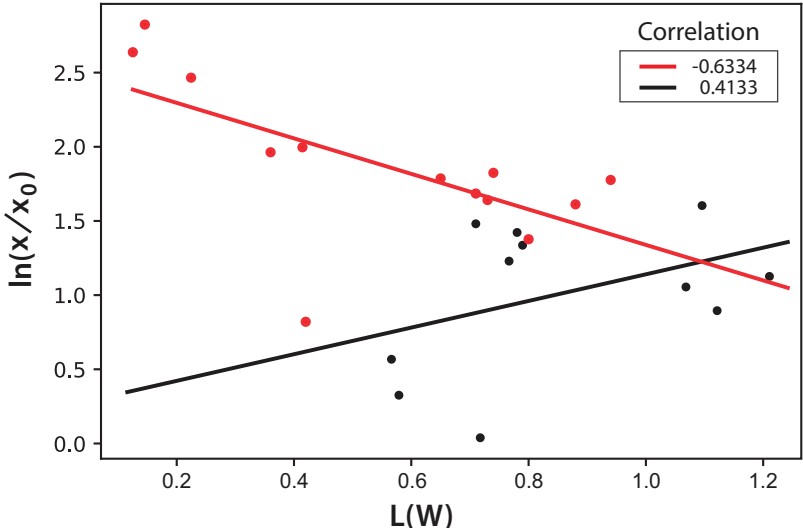

**Figure 9 Logarithmic dependence of cell concentration with the luminosity (L) value in the CIELAB scale of colors for the PBR experiments with light/dark cycles.** The colors correspond to experimental values: black (Exp. 4), red (Exp. 5); the lines correspond to the mean-squares correlations.            

luminosity L for Experiments 1–3. For these experiments, Eq. (3) is proposed to get $x$ from on-line measurements of $L$ from digital images:

$$\ln\left(\frac{x}{x_0}\right) = (1.6 \pm 0.2) - (0.44 \pm 0.04)\,(L/W) \tag{3}$$

**Table 2 $L(W)$ correlation.**

|  | E1 | E2 | E3 |
|---|---|---|---|
| E1 | 1.000000 | 0.963332 | 0.942350 |
| E2 | 0.963332 | 1.000000 | 0.973427 |
| E3 | 0.942350 | 0.973427 | 1.000000 |

**Table 3 $\ln(x/x_0)$ correlation.**

|  | E1 | E2 | E3 |
|---|---|---|---|
| E1 | 1.000000 | 0.973283 | 0.948333 |
| E2 | 0.973283 | 1.000000 | 0.985426 |
| E3 | 0.948333 | 0.985426 | 1.000000 |

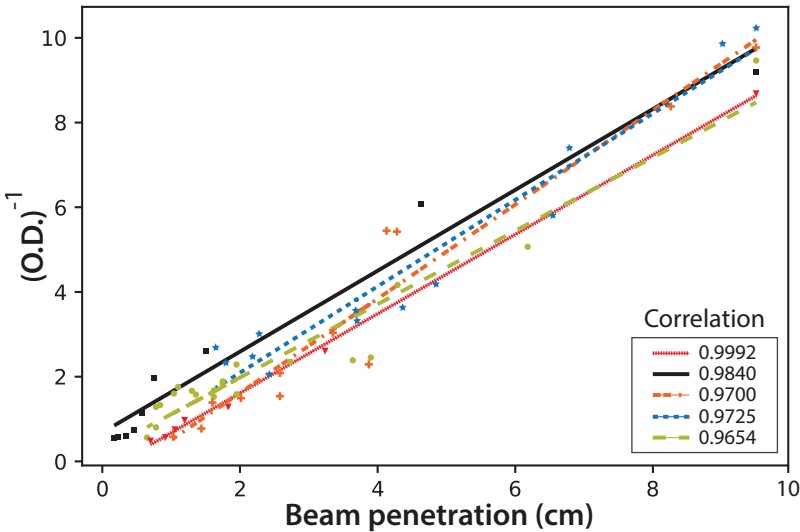

**Figure 10 The inverse of microalgae culture optical density (OD) as a function of the fluorescent beam penetration in the PBR.** The colors correspond to the following experimental values: red dash line (Exp. 1), black (Exp. 2), orange dotted line (Exp. 3), blue dotted line (Exp. 4) and light green dotted line (Exp. 5); the lines correspond to the least-squares correlations.

In this equation, the intercept and the slope values are presented as the mean and the corresponding standard deviation of the individual correlations in the three experiments (see Fig. 8). However, this same correlation was not observed in Experiments 4–5, where light/dark cycles were performed (Fig. 9). The correlations between the values of the three experiments confirm that the experiments are reproducible, which gives confidence to the study. Table 2 shows the $L(W)$ and Table 3 shows the correlation between $\ln(x/x_0)$ values.

Finally, the fluorescence beam penetration was characterized as an image changing with time. Both the beam surface area (data not shown) and penetration distance were used to monitor these changes. Similar results were obtained when comparing the cell concentration with the changes in the fluorescent images; therefore, only the distance

beam penetration was used since its measurement was much simpler than that of the surface area. Figure 10 shows the correlation of this distance measured for the fluorescent beam penetration with the inverse of the OD. As stated before, the OD is related to the biomass concentration (*Del Campo et al., 2014*). It is important to note that the linear correlations were obtained for all experiments, with Eq. (4) proposed to calculate the OD of the culture directly from the on-line measure of the beam penetration:

$$OD = \frac{1}{Beam\ penetration/cm} \tag{4}$$

This simple equation is proposed since the values for the mean and the corresponding standard deviation for the intercept and the slope in the individual linear correlations for the five experiments are $(0.0 \pm 0.5)$ and $(1.0 \pm 0.1)$ cm$^{-1}$ respectively.

## CONCLUSIONS

The growth of *C. reinhardtii* cultures was successfully monitored through off-line and on-line optical techniques at an affordable cost. It was confirmed that, as evidenced in green plants, the maximum fluorescence around $\lambda = 685$ nm occurs before that at $\lambda = 740$ nm. The maximum fluorescence at $\lambda = 740$ nm occurs at a higher concentration, than what is needed at $\lambda = 685$ nm. Once the maximum fluorescence at $\lambda = 685$ nm has been reached, it decreases earlier and at a faster rate than the fluorescence at $\lambda = 740$ nm. Although the $F_{685}/F_{740}$ fluorescence ratio is a well-known method, it was used for *C. reinhardtii* cultures for the first time here. A very useful linear correlation occurs between the logarithmic concentration of *C. reinhardtii* and the $F_{685}/F_{740}$ ratio over time.

Moreover, the on-line analysis of digital images was shown to be useful in monitoring *C. reinhardtii* growth. The luminosity measurements in the CIELAB scale correlated linearly with the microbial concentration for cultures under continuous illumination. However, this correlation was not found for the cultures in a light/dark regime. Nevertheless, both the fluorescent beam penetration images' distance and the surface captured for the beam linearly correlated with OD and, consequently, with the microalgae culture density for all the illumination regimes. A simple reciprocal equation can be used to calculate OD as the inverse of the measured distance of beam penetration ($\lambda = 440$ nm).

The on-line techniques proposed here are very practical for both research and the study of industrial microalgae cultures. In the case of multispectral remote sensing reflectances at both 685 and 740 nm, and with the contribution of field measurements for calibration, a future study should use Eq. (2) and regression models between the logarithmic concentration of microalgae and remote sensing data to estimate the concentration of Chl a in wide water areas.

## ACKNOWLEDGEMENTS

The authors are grateful to the editor and reviewers for improving this manuscript. We thank Moisés Perales for his help with editing.

### Funding

Hugo-Enrique Lazcano-Hernández received a postdoctoral fellowship from CONACYT and the support through the "Cátedras-CONACYT" program (project 526). The funders had no role in study design, data collection and analysis, decision to publish, or preparation of the manuscript.

### Grant Disclosures

The following grant information was disclosed by the authors:
CONACYT and the support through the "Cátedras-CONACYT" program: project 526.

### Competing Interests

The authors declare that they have no competing interests.

### Author Contributions

- Hugo-Enrique Lazcano-Hernández conceived and designed the experiments, performed the experiments, analyzed the data, contributed reagents/materials/analysis tools, prepared figures and/or tables, authored or reviewed drafts of the paper, approved the final draft, redesign of experiments and analysis.
- Gabriela Aguilar performed the experiments, analyzed the data, contributed reagents/ materials/analysis tools, prepared figures and/or tables, authored or reviewed drafts of the paper, approved the final draft.
- Gabriela Antonia Dzul-Cetz performed the experiments, analyzed the data, contributed reagents/materials/analysis tools, prepared figures and/or tables, authored or reviewed drafts of the paper, approved the final draft.
- Rodrigo Patiño conceived and designed the experiments, analyzed the data, contributed reagents/materials/analysis tools, authored or reviewed drafts of the paper, approved the final draft.
- Javier Arellano-Verdejo analyzed the data, prepared figures and/or tables, approved the final draft.

### Data Availability

The raw measurements are available in the Supplemental Files.

### Supplemental Information

Supplemental information for this article can be found online at http://dx.doi.org/10.7717/ peerj.7956#supplemental-information.

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
