# Peer review of "Off-line and on-line optical monitoring of microalgal growth"

_PeerJ, doi:10.7717/peerj.7956_

## Round 0.1 · original submission · Minor Revisions

Please consider the reviewer's comment in their entirety. The references that Reviewer 2 suggests should be included since it is important to examine the work in the context of other related scientific work. The paper is an interesting approach to characterize microalgae cell concentrations in real-time for aqua culture applications and I look forward to receiving your revised manuscript.

·

Basic reporting

First, I would like to commend the authors for their work. It is a very interesting and useful research. Second, I appreciate for the raw data set available. In my opinion the article is good, but it needs an improvement on English language to ensure that it follows the professional clearly written form. Some verb tenses should be fixed to the past, which I think is the right way to describe a research done. There are also some extra commas. I suggest that you have a native English spoken colleague reviewing your manuscript.

I liked the organization of the manuscript, the objective is clear, the methods explained and discussed later. Conclusion is also understandable.

Experimental design

The experimental design it not well explained. The experimental design, the main hypothesis, treatments and how the data were going to be analyzed are not described.

I suggest a better explanation about the statistics analysis.

Validity of the findings

I consider the results valid although the statistic design is not well explained.

Reviewer 2 ·

Basic reporting

“Off-line and on-line optical monitoring of microalgal growth,” by Lazcano-Hernandez et al., describes the implementation of three optical approaches for assessing microalgal cultures based upon (1) the chlorophyll fluorescence ratio F685/F740, (2) the color of a culture backlit by a fluorescent lamp, and (3) the penetration depth of a superluminescent diode. In motivating this work, the authors state that “real time non-invasive methodologies are still needed to monitor the growth conditions of the microalgae culture.” As reviewed by Havlik et al. (“Monitoring of microalgal processes,” Adv. Biochem. Eng. Biotechnol. 153, 89-142, 2016), there have been several such methods pursued over the past decade, and I would recommend that the paper reference this relevant prior work.

Experimental design

As the authors state, the trends they observe in the chlorophyll fluorescence ratio F685/F740 of Chlamydomonas reinhardtii have been previously reported for plants. Also, the relationship reported between luminosity and cell concentration should be expected to be dependent upon the spectral content of the backlighting (e.g., fluorescence lamps vs. LEDs). To allow for reproduction by others, the spectral content of the fluorescence lamps should be reported. Finally, I would recommend that the authors include the results of Experiments 4-5 in Fig. 8, as graphically depicting these results may motivate others to hypothesize reasons for the lack of agreement with Experiments 1-3.

Validity of the findings

The authors have demonstrated algal biomass assessment by measuring the attenuation of light. This is somewhat routine practice by the phycology community in the measurement of OD 750 (see, e.g., the discussion by Griffiths et al., “Interference by pigment in the estimation of micoalgal biomass concentration by optical density,” J. Microbiol. Meth. 85, 119-123, 2011). Lazcano-Hernandez et al. demonstrate methods for assessing light attenuation without needing to extract a sample. A challenge, left unaddressed by the authors, is that the observed light attenuation remains dependent upon the measurement geometry, which would likely be different for every photobioreactor geometry. And because the optical methods described depend upon the culture’s absorption, each new organism would require a new assessment as well.

Related to the topic of culture absorption, the authors should wonder why the slopes in Fig. 9 are all near unity. The optical density of the culture is being measured at 640 nm, while the luminescent diode exhibits peak emission at 440 nm. Chlamydomonas reinhardtii exhibits 2-3 times more absorption at 440 nm than at 640 nm (see de Mooij et al., “Impact of light color on photobioreactor productivity,” Algal Res. 15, 32-42, 2016), so one would expect the lines in Fig. 9 to display slopes significantly greater than unity.

In the Conclusions, the authors claim that their on-line techniques could be extended to remote sensing applications. Without additional caveats, this should be clarified as speculation. Remote sensing systems typically co-locate the transmitter and receiver, which would not allow for the off-axis lighting and backlighting geometries implemented by the authors in this work.

---

## Round 0.2 · Minor Revisions

Thank you for resubmitting your manuscript. The manuscript has addressed the reviewer's scientific concerns, however, it still needs careful editing for English usage and minor punctuation/grammar mistakes. Attached are some suggested edits through p. 3. Please note that the ENTIRE manuscript needs a careful review for grammatical errors and writing in standard English.

---

## Round 0.3 · Minor Revisions

Thank you again for your resubmission. The manuscript is improved but still has a large number of English grammatical and usage errors. Please consult a professional proofreader/scientific editing service before resubmitting.

There are several other areas that can help the manuscript's clarity.

The two sentences in Lines 29-31 should include several references.

Lines 94-96 should be eliminated or worked into the Introduction

The following sentences beginning at Line 107 are not Materials and Methods and should be incorporated into the Introduction. "In a previous 108 study, the dry weight of the microalgae was correlated with the OD of the algae cultures (del Campo et al.,109 2014). Limitations of the OD technique as well as its applications to other species and another culture 110 medium were discussed by Griffiths and co-workers (Griffiths et al., 2011).

---

## Round 0.4 · accepted · Accept

Thank you for your resubmission and patience.

---

## Author Rebuttal · Round 0.4

# Off-line and on-line optical monitoring of microalgal growth

**Dr. Scott Wallen**
*Academic Editor, PeerJ*

**Dear Dr. Wallen**
We deeply appreciate the suggestions to our manuscript "**Off-line and on-line optical monitoring of microalgal growth**". All suggestions were followed. We present the changes as follow (in blue):

**Thank you again for your resubmission. The manuscript is improved but still has a large number of English grammatical and usage errors. Please consult a professional proofreader/scientific editing service before resubmitting.**

A professional proofreader/scientific editing service was consulted (PeerJ editing service).

**The two sentences in Lines 29-31 should include several references.**

References were included:

**Line 29**
"...(Pulz and Gross, 2004; Spolaore et al., 2006; Mata et al., 2010; Niccolai et al., 2019)..."

**Lines 30-31**
"...(Wang et al., 2008; Abdel-Raouf et al., 2012; Show et al., 2019)..."

**Lines 94-96 should be eliminated or worked into the Introduction**

The sentence was relocated at the end of the "Introduction" section.

**The following sentences beginning at Line 107 are not Materials and Methods and should be incorporated into the Introduction. "In a previous 108 study, the dry weight of the microalgae was correlated with the OD of the algae cultures (del Campo et al.,109 2014). Limitations of the OD technique as well as its applications to other species and another culture 110 medium were discussed by Griffiths and co-workers (Griffiths et al., 2011).**

The sentences were relocated into the "Introduction" section.

**Lines 216-217**
"..., and a final measurement on the seventh day..." was deleted.

**Editing service suggestions.**

**Line 13**
The symbol "," was deleted.

**Line 15**
"...It is shown..." was replaced with "...we found..."

**Line 16**

"...Moreover, with…" was replaced with "...By using..."
"...was…" was erased.

**Line 19**
"...To monitor..." was replaced with "..for…"

**Lines 21-22**
"...transform a fraction of the solar energy to biochemical energy in order to produce their own food..." was updated with "...produce their own food by transforming a fraction of solar energy to biochemical energy..."

**Line 29**
"...microalgae's..." was replaced with "...its…"

**Lines 30-31**
"...the cleaning of residual water..." was replaced with "...residual water cleaning..."

**Line 31**
"...products, which contribute…" was replaced with "...contributors..."

**Line 32-34**
"... A wide review of on-line and off-line technologies to monitor physicochemical and biological parameters of microalgae was compiled by Havlik and co-workers..." was updated with "...Havlik and colleagues compiled a wide review of on-line and off-line technologies to monitor physicochemical and biological parameters of microalgae..."

**Line 34**
The word "also" was added between "are" and "several"

**Line 39-40**
"...and this prevents…" was replaced with "..., preventing…"

**Line 45**
A letter "n" was added in the word "harvesting"

**Line 46-47**
"...their reaction centers' characteristics…" was replaced with "...the characteristics of their reaction centers..."

**Line 53**
"," was added after "temperature".

**Line 61**
"... , however…" was replaced with "... . However…"

**Line 63**
"... . These...." was replaced with "...including:..."

**Line 67-68**
"...monitor their physiological state, including changes in..." was updated with "...monitor changes in their physiological state, such as in…"

**Line 73**
"...an approximate determination of Chl content in a non-destructive way using the Chl ratio…" was updated with "...an approximate non-destructive determination of Chl content using the Chl ratio…"

**Line 77-80**
"...At present, two Chl fluorescence approaches are used to monitor photosynthetic efficiency in microalgae mass cultures. These are rapid fluorescence induction and the saturation-pulse method (Masojídek et al., 2011), which are well known successful methods…" was updated with "...Presently, two Chl fluorescence approaches are successfully used to monitor photosynthetic efficiency in microalgae mass cultures: rapid fluorescence induction and the saturation-pulse method (Masojídek et al., 2011)..."

**Line 85-86**
"...methodos are proposed in this study in order to measure Chlamydomonas reinhardtii (C. reinhardtii) culture growth…." was replaced with "...methods to measure Chlamydomonas reinhardtii (C. reinhardtii) culture growth are proposed in this study…."

**Line 87-90**
"...C. reinhardtii is considered one of the most promising eukaryotic H2 producers (Torzillo et al., 2015), which is why its study is relevant. The methodologies proposed here were applied to study the growth of C. reinhardtii. It is possible to apply the proposed methods to other species. However, results..." was updated with "...The study of C. reinhardtii here is relevant since it is considered one of the most promising eukaryotic H2 producers (Torzillo et al., 2015). It is possible to apply the proposed methods to other species but results..."

**Lines 94-96**
The paragraph was located at the end of the "Introduction" section
**Line 97**
"...were..." was added after "and"

**Line 103**
"...use..." was replaced with "...use**d**…"

**Line 106**
The word "the" after "monitor" was erased.

**Line 108**
"...the microalgae was…" was updated with "...microalgae…"

**Line 109**
A comma was added after "technique"

**Line 110**
A comma was added after "medium"

**Line 112**
"..., with continuous…" was replaced with "... and continuous…"

**Line 117**

"...difference being…" was replaced with "...difference **between them** being…"
The word "...as…" was erased.

**Line 119**
"...Therefore, the number…" was updated with "... and the number…"

**Line 130**
"...These were…" was updated with "...These **lamps** were…"

**Line 130-131**
"...For the fluorescent images, the pictures were captured in darkness and the culture only…" was updated with "...The fluorescent images, were captured in darkness and only the culture…"

**Line 135**
The word "the" was deleted after "and".

**Lines 148-149**
"...Parallel to the floor, as exciting radiation, six LEDs were placed (Steren, Ultra Blue),..." was updated with "...As exciting radiation, six LEDs (Steren, Ultra Blue) were placed parallel to the floor,..."

**Lines 154-155**
"...Traditionally, fluorescence is measured through the sample cuvette wall. Instead of that,..." was updated with "...Fluorescence traditionally is measured through the sample cuvette wall, but our…"

**Lines 157-158**
"...the material and the geometry of the cuvette are not important…" was replaced with "...the cuvette's geometry is not important…"

**Line 158**
"By" was replaced with "using"

**Line 160**
"...; then, the…" was updated with "..., then…"

**Line 175**
The word "the" was deleted before "younger"

**Lines 175-176**
"... with a not developing photosynthetic apparatus…" was replaced with
"... with a photosynthetic apparatus that isn't developing…"

**Line 177**
"...the..." was replaced with "...a…"

**Line 178**
"...through…" was replaced with "...by…"

**Line 179**
"...can be successfully expressed by…" was uploaded with

"...is successfully expressed in…"

**Line 182**
"This" was replaced with "Our"

**Line 197**
"... file (S1)..." was updated with
"... (File S1)..."

**Line 208**
"... than at λ=740 nm…" was replaced with
"... that of λ=740 nm…"

**Line 209**
"are" was replaced with "were"

**Line 211**
"Increase" was replaced with "increase**d**"

**Line 214**
"... it was possible to observe…" was replaced with "...we observed…"

**Lines 217-218**
"...As time went by, the F685/F740 fluorescence ratio decreased, and that means that the photosynthetic processes were improved…" was updated with
"... Over time, the F685/F740 fluorescence ratio decreased, meaning that the photosynthetic processes had improved…"

**Line 222**
"..., which means that the conditions are appropriate…" was updated with
"... and the conditions were appropriate…"

**Line 227**
"...Finally, equation 2…" was replaced with "...Equation 2…"

**Line 232**
"... It was possible to observe that cultures get darker with time…" was updated with
"... We observed that cultures got darker with time…"

**Line 233**
"...This prevents the passage…" was updated with
"...preventing the passage…"

**Figure 3**
"Throughout" was replaced with "over"

**Line 241-242**
"...In fact, luminosity is what diminishes importantly during cell growth since cells deflect or shadow light sources. In Figure 8, it is possible to observe…" was replaced with
"... and luminosity is what more importantly diminishes during cell growth since cells deflect or shadow light sources. Figure 8 shows…"

**Line 248**
"...A calculation of the correlations between the values of the three experiments confirms that…" was updated with
"...The correlations between the values of the three experiments confirm that..."

**Line 254**
The word "is" was replaced with "was"

**Figure 7**
"...Fluorescent bea**n**..." was replaced with "...fluorescent bea**m**…"

**Line 255**
"...In Figure 10, it is possible to observe…" was replaced with
"...Figure 10 shows…"

**Line 256**
"... is already related with…" was replaced with
"... is related to…"

**Line 257**
The word "notice" was replaced with "note"

**Line 266**
The word "happening" was deleted.

**Line 267**
"..., compared to…" was replaced with "...than…"

**Line 268**
"...decreases before and at a faster rate that the…" was updated with
"...decreases **earlier** and at a faster rate tha**n** the…"

**Lines 269-270**
"..., here it was demonstrated for the first time for C. reinhardtii cultures…" was replaced with
"... it was used for C. reinhardtii cultures for the first time here…"

**Line 271**
"through" was replaced with "over"

**Line 272**
"...was also shown to be useful to monitor…" was replaced with
"...was shown to be useful in monitoring…"

**Line 273**
"...were linearly correlated with…" was replaced with
"...correlated linearly with…"

**Lines 274-276**

"...for the cultures in a light/dark regime, this correlation was not found. Nevertheless, for the fluorescent beam penetration images, both the distance and the surface captured for the beam were linearly…" was replaced with

"...this correlation was not found for the cultures in a light/dark regime. Nevertheless, both the fluorescent beam penetration images' distance and the surface captured for the beam linearly…"

**Line 277**
"Indeed" was erased.

**Line 280**
"...to study both research and industrial microalgae…" was replaced with
"... for both research and the study of industrial microalgae…"

**Line 281**
"...As a future study, in the case of having multispectral remote sensing reflectances both at 685…" was replaced with
"...In the case of multispectral remote sensing reflectances at both 685…"

**Line 282**
"...it would be feasible to use…" was replaced with
"...a future study should use…"

**Line 288**
"...their contribution to improve…" was replaced with
"...improving…"

**Line 289**
"...the manuscript…" was erased.

Best Regards.
The authors.